# Application of the DEA Model in Tourism SMEs: An Empirical Study from Slovakia in the Context of Business Sustainability

**Ján Dobrovič** [1,*], **Veronika Čabinová** [2,*], **Peter Gallo** [3], **Petra Partlová** [1], **Jan Váchal** [1], **Beata Balogova** [3] **and Jozef Orgonáš** [4]

[1] Department of Management, Faculty of Corporate Strategy, Institute of Technology and Business in České Budějovice, Okružní 517/10, 370 01 České Budějovice, Czech Republic; partlova@mail.vstecb.cz (P.P.); vachal@mail.vstecb.cz (J.V.)

[2] Department of Development, Informatization and Quality Assessment, University of Prešov in Prešov, 080 01 Prešov, Slovakia

[3] Institute of Educology and Social Work, Faculty of Arts, University of Prešov in Prešov, 080 01 Prešov, Slovakia; peter.gallo.1@unipo.sk (P.G.); beata.balogova@unipo.sk (B.B.)

[4] Faculty of Commerce, University of Economics in Bratislava, Dolnozemská Cesta 1, 852 35 Bratislava 5, Slovakia; jozef.orgonas@euba.sk

\* Correspondence: jan.dobrovic1@gmail.com (J.D.); veronika.cabinova@unipo.sk (V.Č.)

**Abstract:** Slovak spa services are not given sufficient attention directly following the support and sustainable development. The paper focuses on the evaluation of the overall development and current level of efficiency of the Slovak spas in 2013–2018, through the application of DEA models. Input variables (total number of beds, employees, medical staff) and output variables (use of bed capacity, number of treated clients) within the structure of DEA models analyzed (CCR-I, CCR-O, BCC-I, BCC-O) are determined by results of the correlation analysis. The data were obtained from the annual reports of the spa enterprises. By the results, the average efficiency score for all enterprises reached 0.7527, i.e., the average spa enterprise would need only 75.27% of currently used inputs for a given output production to move to the efficiency frontier. The development of the average efficiency score confirmed a positive growing trend until 2015; however, the efficiency decreased by 1.84% in a year-to-year comparison in 2016–2018. In each year of the analyzed period, the number of inefficient enterprises (66.67%) exceeded that of the efficient ones (33.33%). Through research carried out in spa facilities, the authors contributed to expanding the application of the DEA method in another tourism sector.

**Keywords:** data envelopment analysis; business sustainability; SMEs; efficiency; benchmarking; spa enterprises

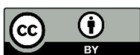

## 1. Introduction

Due to changing global business environment, the issue of evaluating business efficiency is currently discussed more, not to mention the effort to develop new, modern approaches to its solution. Due to the development of statistical methods and the growing interest of financial institutions and the enterprises, there is a significant increase in the use of innovative methods and the models for evaluating business efficiency in various economic sectors, particularly in the past decade. It also applies to the progressively evolving field of tourism, as its support and balanced sustainable development began to play a crucial role in the national economies [1]. The research sample is based on current demographic trends, using a specific and attractive research sample—the spa enterprises. Spa tourism is one of the most popular forms of recreation in the modern world. In the last three decades, there has been a dynamic development of spa tourism and other forms of health tourism, such as wellness and medical tourism [2,3]. By the medium variant of

the United Nations forecast calculated for the countries of the European Union, it is expected that almost 29% of the population will be over 65 years of age by 2050, providing space for promising clients. Compared to other European Union Member States, there is a growing interest in business activity in Slovakia [4]. The business sector is undoubtedly one of the most important parts of today's market economy. SMEs produce and sell the major part of goods and services, thus contributing to the sales tax revenues, GDP and job creation. In this regard, their role in the economy is irreplaceable and justified [5]. However, during pandemic, enterprises face pressure and are forced to deal with many unexpected problems and challenges in order to ensure sustainable entrepreneurship [6]. Undoubtedly, tourism is the engine of economic growth, as reflected in foreign exchange earnings; contributions to private and public revenues; in job creation; incentives for technology development; in the formation of human capital, business opportunities, etc., [7–9]. To the above, together with other reasons, it is therefore important to pay attention to such form of tourism, assess its current state, and focus on its support and sustainable development.

The main goal of the paper is to evaluate the overall level of efficiency of the spa enterprises operating in Slovakia, using the DEA models. The intention is based on a quantified efficiency score to compile a rank rating of the spa enterprises, identify their strengths and weaknesses, identify the industry leaders (benchmarks), process a projection of reference values of input variables leading to the required efficiency limit and thus contribute to improving its management in the context of sustainable development of the spa treatment in Slovakia.

The structure of the paper is as follows. In the second section, there is a review of the professional studies carried out in the field of business efficiency evaluation. Special attention is paid to the methods of the evaluation with a focus on data envelopment analysis (DEA), applied in the research part of the paper. The third section of the paper contains a description of the research sample of the enterprises, data sources and applied methods. In the following section, supported by relevant research studies, four different DEA models were practically applied and compared. Subsequently, based on the most appropriate, input-oriented DEA model with the assumption of variable returns from the scale (BCC-I), the overall efficiency of the Slovak spa sector is analyzed, also focusing on analyzing the level of efficiency of different spa enterprises. Last two sections deal with the discussion, the comparison of the achieved results with comparable empirical studies, the limits of the paper, the future scientific direction, and also the overall summary of the results.

Since the literature review confirmed the assumption that no research study analyzing the level of efficiency of the spa sector and the enterprise operating in it has been carried out in Slovakia so far, it seemed necessary to carry out the research in this area and contribute to deepening the current knowledge. The compilation of a practical application of the DEA model can serve as a basis for further analysis and definition of efficiency and performance indicators. The model can be used not only in the spa sector, but also in other areas of the economy with the adjustment of the relevant indicators.

## 2. Literature Review

The following part summarizes the results and knowledge based on literature review related to the analysis of business efficiency. Such area plays an important role in the conditions of a tough competitive environment, as its constant monitoring ultimately helps in the managing and increasing the performance of the enterprise [10–12]. Current enterprises need to respond to the ever-changing market situation, adapt to new changes, and improve business activities, essential for the sustainability of their future growth [13–17].

Some authors [18–23] notice that the enterprises invest a lot of financial and non-financial resources to gain a competitive advantage, ensuring a high level of corporate performance and long-term sustainability. For such reason, the issue of quantification and subsequent management of efficiency are at the heart of a growing number of discussions,

which, however, face many challenges with the need to be addressed without delay [24–28].

### 2.1. Corporate Efficiency–Theory Background

Efficiency is an important criterion of evaluating business results, one of the main goals of the implementation of economic and financial activities of the enterprise. The definition of efficiency is quite challenging, as there are many opinions and approaches to the issue. Some authors [29–32] define business economic efficiency. By those authors, efficiency consists of two basic components, namely *technical efficiency*, reflecting the ability to achieve maximum output from the set of inputs and *allocation efficiency*, reflecting the ability to use inputs in optimal proportions with respect to their prices and production technology. Pritchard [33] defines efficiency as the degree to which the enterprises use their limited resources to produce final products and services. The term also refers to the relation of outputs to certain standards and expectations. Azimi [34], Bulinska-Stangrecka and Bagieńska [35] define efficiency as a functional characteristic of business activity. It expresses the overall rationality of its activities as a dedicated system that works only on the basis of purposefully secured links with the environment [36]. Trivedi [37] defines business efficiency as the degree to which an enterprise rationally allocates its limited resources to achieve predetermined goals after taking the constraints of the internal and external environment into account. By Callender [38], Repnikova et al. [39], Nikonorova et al. [40] efficiency is described as a purposeful process of meeting the ever-growing needs of the society at the maximum possible level.

The basis of the concept of the efficiency is the "effect" in terms of a result and a consequence. A common effect for all the enterprises is related to the products and services provided referred to as the outputs, which are the result of the consumption of production factors that make up the inputs. Based on the above, efficiency is expressed as the ratio of outputs to inputs. Economic theory defines efficiency as a state where it is not possible to produce another product or service at a given resource without having to limit the production of another product or service. The production unit thus moves to the limit of production possibilities, which does not lead to waste. In the real world, it is necessary to accept the assumption of the existence of waste, as in the current market economy there are efficient and inefficient production units [41].

The more efficiently an enterprise works, the more efficient it is in the implementation of its production on national and global markets. Achieved higher efficiency helps the enterprise in better, and cheaper implementation of its strategic activities, compared to the competitors, which in turn leads to gaining a competitive advantage and improve the sustainability of the enterprise [42]. The key to improving efficiency is the analysis of mechanisms in the case of specific factors affecting efficiency and taking measures in an orderly manner depending on the effects [43]. Pakhnenko et al. [44] notice that the management of sustainable development of the enterprises requires the improvement of methodological approaches to their evaluation of effectiveness. The problem of assessing the economic efficiency of the corporate activities therefore lies primarily in the definition of appropriate criteria, and approaches, as addressed in the following subchapter.

### 2.2. Approaches to Measuring Business Efficiency

Measuring efficiency, and identifying the sources of potential inefficiencies, is rather an important step in improving the competitive position of the enterprises and their overall behavior in a competitive environment. At present, there is a wide range of methods and procedures by which it is possible to measure the efficiency of the activities of production units. The following reports the most important approaches to quantifying efficiency:

- The ratios—by [44], the ratios are the most common method of efficiency evaluation, as their relatively simple quantification is based on the current financial statements.

Their biggest drawback is that they focus only on a limited number of factors that do not have a sufficient impact on the overall efficiency of the production unit. However, they are useful for the basic orientation of the operation of the monitored unit. For a more detailed analysis of efficiency, it is then necessary to use more complex tools of economic analysis based on mathematical modelling.

- Parametric methods—a group of parametric methods is stochastic in nature, i.e., they contain at least one random component. The aim of the methods is to distinguish inefficiency from the effects of random errors, related to a higher reliability of the final results. Their disadvantage is that the given methods define a specific functional dependence, which determines the shape and course of the efficiency limit. If these assumptions do not correspond to reality and the functional dependence is not defined correctly, the final results may be damaged by specific errors and the final results are distorted. The methods quantify economic efficiency, such as stochastic frontier approach, distribution free approach, thick frontier analysis, corrected ordinary least squares.
- Nonparametric methods—a group of nonparametric methods is of a deterministic nature, i.e., they do not contain any random component. Therefore, it is not possible to effectively eliminate the negative consequences of accidental errors, measurement errors or incomplete data in the quantification of efficiency. With these methods, the assumptions for production technology are not as strict as with parametric methods, therefore a higher degree of freedom is permissible for the examined units. Compared to parametric methods, this group quantifies not economic but technical efficiency. The group includes methods such as DEA, free disposal hull, stochastic data envelopment analysis [45].

As reported by the above-mentioned authors, the parametric approaches are generally regression techniques that assume the existence of a special functional form for a boundary and thus determine the inefficiency against this criterion. Nonparametric methods evaluate the inefficiency relative to all units in the sample. The most important difference between deterministic and stochastic methods is their attitude to the random component. The deterministic approaches assume that any deviation from the border is caused only by inefficiency. On the other hand, the stochastic approaches attach weight to the deviation from the border not only of inefficiency, but also of the existence of randomness. The results are comparable when applying both approaches, but there may be minimal differences. However, other methods are often used in the literature to measure efficiency (mostly economic), such as mathematical programming, econometrics, and simulation methods.

### 2.3. Data Envelopment Analysis and Its Use

At the most general level of understanding, the DEA method is used to quantify the technical efficiency of comparable production units producing certain outputs, for the production of which they consume certain inputs. The units are, for example, school facilities, hospitals, banking institutions, public and state administration facilities, national economies, and economic sectors.

The initial ideas of assessing technical efficiency is traced back to the second half of the 20th century. Debreu 1951 [46] developed a basic methodology for analyzing the technical efficiency of units. This methodology was able to accept several input variables, creating a generally applicable and comprehensive measure of efficiency. The author's approach was modified a few years later for the case of multiple outputs and formulated as a problem of linear programming by [47]. Since the above-mentioned authors introduced the DEA method to the world, it has become a popular subject of research in many empirical studies, and its popularity has continued to grow progressively in recent years. However, its applicability is not limited exclusively to the analysis of the efficiency of the production units, the scientists are increasingly looking for its application in other areas.

A review of world empirical studies concerning the DEA method and trends in its future development is reported by [48]. Due to the fact that the DEA method is the subject of research of a huge number of research studies, this paper focuses on the studies carried out in areas related to spa treatment, as much as possible (especially the course and results of the production process).

By Dénes et al. [49], in order to introduce the application of the DEA method in various areas of economic life, it was necessary to define a designation for the analyzed unit, within which the inputs are transformed into the outputs. The term *Decision Making Unit* was thus introduced. The applicability of the DEA method is justified and correct only if all DMUs perform the same or similar activity. Only then is it possible to identify a common group of inputs and outputs that are relevant to the analysis. Let us suppose that there is a set of homogeneous production units: $U_1$, $U_2$, …, $U_n$. In measuring the efficiency of the enterprises, each of the units produces r of the outputs and at the same time consumes m of the inputs. Then, $X = \{x_{ij}, i = 1, 2, …, n; j = 1, 2, …, m\}$ is the input matrix and $Y = \{y_{ik}, i = 1, 2, …, n; j = 1, 2, …, r\}$ is the output matrix. By [41], the efficiency of the unit is generally expressed by the following:

$$\text{eff}(U_q) = \frac{\text{weighted sum of outputs}}{\text{weighted sum of inputs}} = \frac{\sum_{k=1}^{r} u_k\, y_{qk}}{\sum_{j=1}^{m} v_j\, x_{qj}} \tag{1}$$

In the formula, $v_j$, $j = 1, 2, …, m$ are the weights matching the j-th input and $u_i$, $k = 1, 2, …, r$ are the weights matching the k-th output. DEA models maximize the efficiency measure of the analyzed unit $U_q$, expressed as the ratio of the weighted outputs and the weighted inputs, provided that the efficiency rates of all other units are less than or equal to one. Input and output weights must be greater than zero at the same time to include all considered characteristics in the model.

There are different DEA models, based on different classifications. The *input-oriented DEA models* analyze the efficiency of the enterprises based on the input variables. The enterprises with optimal value of the purpose function reaching one are considered efficient. With a value of less than one, the enterprises operate inefficiently, reporting the need to reduce inputs in such a way that an inefficient enterprise becomes efficient. The *output-oriented DEA models* answer the question, to what extent the outputs should be increased without changing the level of inputs, i.e., they perceive efficiency as the ability to produce the maximum number of the outputs for a given input. Due to the limited scope of the paper, sample empirical studies dealing with the application of input and output-oriented DEA models are discussed, exclusively in the field of hospitality and healthcare, as the authors of the paper analyze the spa facilities as a special type of enterprise, that have not been the subject of any efficiency studies using the DEA models so far. It is assumed, that their activities are included in both of the above-mentioned areas, although priority attention should be paid mainly to the provision of spa health care. The following authors applied input-oriented DEA models in hotel facilities in their studies [50,51]. Many authors [52–56] deal with health-care facilities. The application of output-oriented DEA models in hotel facilities is under discussion [57] as well as the application of models in healthcare facilities [58–61].

In general, the most important DEA models also include the CCR model [62], formed as an acronym for the surnames of its authors (sometimes referred to as the CRS model). The model assumes constant returns to scale, and it expects that the change in outputs/inputs is the same as the change in inputs/outputs. There are also the models based on the assumption of variable returns from the scope, such as the BCC model [62], which is essentially a modification of the CCR model and its name is also an acronym for the surnames of its authors. The model assumes that the level of outputs/inputs does not have to change in the same proportion as the level of inputs/outputs—it can increase, decrease, and remain constant. In terms of the form of technical in/efficiency, which the model is able to achieve with its degree of efficiency, the BCC model is classified as a radial DEA model. In the areas of hospitality and healthcare the authors analyzed, which DEA models

(CCR or BCC) are used most often. In the hotel industry, the application of the CCR DEA model is preferred by [63]. On the other hand, the use of the BCC DEA model is considered more appropriate by the authors [64–66]. Models (CCR and BCC) subsequently investigated possible deviations. Regarding health-care facilities, Lo Storto and Goncharuk [67] prefer the application of the CCR DEA model. On the other hand, Sendek et al.[68] consider the use of the BCC DEA model to be more relevant. As in the case of hotel facilities, also in the facilities providing primary health care, Lacko et al. [69], Papadaki and Staňková [70], Szabo et al. [71] report that it is more appropriate to apply DEA models with constant and variable returns to scale and subsequently analyze and compare the results of the achieved efficiency score in detail.

Radial DEA models reflect the degree of efficiency pointing to the need for proportional reduction of inputs, and the expansion of outputs so that the unit becomes efficient—e.g., CCR model, BCC model, and radial DEA models for calculation of super-efficiency. Non-radial DEA models explore the possibilities of disproportionate changes in inputs and outputs in order to achieve efficiency, such as the model of [72], the SBM model of [73]. It is a model deriving technical efficiency from the size of input and/or output chutes, depending on the orientation. The author of this model also summarizes its basic properties, which are the scale invariance with respect to the units of measure used and also the fact that the optimal solution of the SBM model monotonically decreases with the increase of each slip in the inputs and outputs.

## 3. Materials and Methods

The main goal of the paper is to evaluate the overall level of efficiency of the spa enterprises operating in Slovakia, based on the application of the DEA models. The intention is based on a quantified efficiency score to compile a rank rating of the spa enterprises, identify their strengths and weaknesses, identify the industry leaders (the benchmarks), process projections of input variables leading to the required efficiency limit, and thus contribute to improving its management in the context of sustainable development of the Slovak spa treatment.

### 3.1. Description of the Research Sample

The research sample consisted of the spa enterprises operating in Slovakia, which are, with regard to the current demographic development of the population, considered to be the main product line of tourism. By Szromek [74], Gavurova et al. [75] the spa treatment is a very specific tourist unit that performs tasks in the field of tourism (in the private sector) and public health (in the public sector).

By the statistical classification of economic activities of the Statistical Office of the Slovak Republic, the spa enterprises are classified under the section Q—Human health and social work activities; group 86—Human health activities, classification 86.909—Other human health activities.

Currently, in terms of the legal form of business in the field of spa care, there are a total of 18 joint-stock companies, three limited liability companies, three contributory organizations, two state-owned enterprises and two non-profit organizations. Most of these spas are in national private ownership (64.29%), followed by state ownership (25%), international private ownership (7.14%), and one spa facility is owned by associations, political parties, and churches. However, nine spa enterprises were excluded from the analysis due to non-established cooperation and also the fact that there were several non-profit and contributory organizations in the area, which could not be included in the analysis due to the fundamental peculiarities of funding and the legal framework of the Slovak Republic. The resulting research sample consisted of a total of 21 spa companies: BARDEJOVSKÉ KÚPELE, Inc. (SE01), HOREZZA, Inc. (SE02), Kúpele Bojnice, Inc. (SE03), Kúpele Dudince, Inc. (SE04), KÚPELE LUČIVNÁ, Inc. (SE05), KÚPELE LÚČKY Inc. (SE06), Kúpele Nimnica, Inc. (SE07), KÚPELE NOVÝ SMOKOVEC, Inc. (SE08), KÚPELE SLIAČ, Inc. (SE09), KÚPELE ŠTÓS, Inc. (SE10), Kúpele Trenčianske Teplice, Inc. (SE11),

KÚPELE VYŠNÉ RUŽBACHY, Inc. (SE12), Liečebné termálne kúpele, Inc. (SE13), Prírodné jódové kúpele Číž, Inc. (SE14), SLOVENSKÉ LIEČEBNÉ KÚPELE PIEŠŤANY, Inc. (SE15), Slovenské liečebné kúpele Rajecké Teplice, Inc. (SE16), Slovenské liečebné kúpele Turčianske Teplice, Inc. (SE17), Kúpele Horný Smokovec, Ltd. (SE18), PIENINY RESORT, Ltd. (SE19), SLOVTHERMAE, Kúpele Diamant Dudince, s.e. (SE20), Špecializovaný liečebný ústav Marína, s.e. (SE21).

### 3.2. Data and Methods

In the process of obtaining, collecting, and processing information and data, the generally known and widespread methods of scientific research are used, such as the methods of analysis, synthesis, deduction, induction, comparison, description, analogy, and descriptive statistics. Data envelopment analysis (DEA) is applied in order to quantify the degree of efficiency of the sample of spa enterprises. In the case of the correct selection of input and output variables within the DEA model, it was necessary to examine their interdependence and remove all the duplications, and choose a more suitable input/output, for which the correlation analysis is used. Using benchmarking, based on the achieved efficiency score, the location of the spa enterprises in the industry was analyzed, an overview of their strengths, weaknesses, and trends was gained and an opportunity was offered to them to take relevant measures to achieve a higher level of efficiency.

Secondary sources as reported by the financial statements of the enterprises and the annual reports are processed in the MS Excel program. The non-financial input and output variables, not mentioned in the annual reports, were reported through additional e-mail communication with the employees of the top management of the enterprises. The excel-coded data are processed in the R programming language (version 3.6.1) using mathematical-statistical methods. The DEA Solver programme (LV 8.0.) is used to calculate the efficiency of the enterprises.

### 3.3. Construction of the DEA Model and the Input/Output Variables

Depending on the analyzed industry, it is possible to apply several DEA models, while the choice of the resulting model is influenced by many factors such as the size of the DMU, the nature of the DMU, the availability of data, the evaluation criteria or the very purpose of carrying out the necessary analysis. The most important thing when choosing a DEA model is to choose the correct orientation of the model (input/output-oriented models, non-oriented models) and the form of (in) efficiency that they are able to contain (radial/non-radial models).

Thus, the first important decision in specifying the use of a suitable DEA model is to choose from *input, output-oriented, or non-oriented models*. Choosing the final DEA model is based the studies carried out in the field of hospitality and healthcare, as mentioned above. In our opinion, the activities of the Slovak spa enterprises focus primarily on providing spa care in order to improve the health of the clients, so more attention is paid to empirical studies related to healthcare, although in the case of hospitality the results are not significantly different. The controllability of inputs is more permissible and realistic in the area compared to outputs—despite the fact that reducing inputs in the provision of health care is undesirable in principle and the demand for health care services tends to increase, not decrease. However, it is important to point out that in this case the outputs are beyond the control of the production unit and the potential manager is able to regulate mainly the input variables.

The second important decision within the specification of the use of a suitable DEA model is the choice between *the radial and non-radial models*. The purpose of each research study is specific; so many authors are inclined to apply the basic CCR model and subsequently the modified BCC model. Differences in quantified efficiency are often minimal. In the field of spa care, however, based on the analysis of empirical studies, the application of the BCC model is more suitable, as it is not possible to assume constant returns from

scale in terms of linear increase of outputs with increasing inputs and vice versa. The assumption of constant returns to scale can only be accepted if all DMUs operate at the optimum size.

From several applied DEA models, after studying the research studies carried out in the field of healthcare and hospitality, an input-oriented DEA model with variable scale returns (BCC-I) was used. Nevertheless, other most common models (CCR-I, CCR-O, BCC-O) were applied too. Table 1 reports the initial mathematical formulation.

**Table 1.** Mathematical formulation of basic DEA models. Source: Adapted with permission from Dlouhý, M.; Jablonský, J.; Zýková, P. (2018) [41].

| DEA Model | Mathematical Formulation | | |
|---|---|---|---|
| CCR-I input-oriented model | Maximize | $\sum_{i=1}^{r} u_i\, y_{iq}$, | |
| | Subject to | $\sum_{j=1}^{m} v_j\, x_{jq} = 1,$ $\sum_{i=1}^{r} u_i\, y_{ik} \leq \sum_{j=1}^{m} v_j\, x_{jk},$ $u_i \geq \varepsilon,$ $v_j \geq \varepsilon,$ | $k = 1, 2, \dots, n,$ $i = 1, 2, \dots, r,$ $j = 1, 2, \dots, m.$ |
| CCR-O output-oriented model | Minimize | $\sum_{j=1}^{m} v_j\, x_{jq}$, | |
| | Subject to | $\sum_{i=1}^{r} u_i\, y_{iq} = 1,$ $\sum_{i=1}^{r} u_i\, y_{ik} \leq \sum_{j=1}^{m} v_j\, x_{jk},$ $u_i \geq \varepsilon,$ $v_j \geq \varepsilon,$ | $k = 1, 2, \dots, n,$ $i = 1, 2, \dots, r,$ $j = 1, 2, \dots, m.$ |
| BCC-I input-oriented model | Maximize | $\sum_{i=1}^{r} u_i\, y_{iq} + \mu$, | |
| | Subject to | $\sum_{j=1}^{m} v_j\, x_{jq} = 1,$ $\sum_{i=1}^{r} u_i\, y_{ik} + \mu \leq \sum_{j=1}^{m} v_j\, x_{jk},$ $u_i \geq \varepsilon,$ $v_j \geq \varepsilon,$ $\mu\text{-random}$ | $k = 1, 2, \dots, n,$ $i = 1, 2, \dots, r,$ $j = 1, 2, \dots, m,$ |
| BCC-O output-oriented model | Minimize | $\sum_{j=1}^{m} v_j\, x_{jq} + v$, | |
| | Subject to | $\sum_{i=1}^{r} u_i\, y_{iq} = 1,$ $\sum_{i=1}^{r} u_i\, y_{ik} \leq \sum_{j=1}^{m} v_j\, x_{jk} + v,$ $u_i \geq \varepsilon,$ $v_j \geq \varepsilon,$ $\mu\text{-random}$ | $k = 1, 2, \dots, n,$ $i = 1, 2, \dots, r,$ $j = 1, 2, \dots, m,$ |

### 3.4. Selection of Input and Output Variables

In the case of the DEA method, there is no general procedure for selecting the appropriate input and output variables from the available data set into the model. Despite the efforts of some researchers to incorporate statistical methods into their decision-making, there is still some ambiguity in the available literature. Therefore, the empirical studies in the hotel and healthcare sector carried out are used as a starting point for the selection of inputs and outputs within the DEA model used in the paper, based on which the following variables are considered in choosing the inputs—total number of beds, total number of employees, total number of medical staff, number of doctors, number of nurses, number of physiotherapists, number of assistants, amount of material costs, amount of operating costs, amount of wage costs, amount of total costs per a bed, average time of hospitalization. The possibility of using these variables as the outputs is considered—the number of treatment days, the number of treated clients, the use of bed capacity, the amount of net profit per a doctor, the number of outpatient visits.

As the DEA method is sensitive to the missing values in the sample, it is important to assess the fulfillment of the condition of data completeness before performing further

analyses. However, some spa facilities did not keep any records of the data as requested, so it was necessary to omit the following variables—the number of nurses, the total cost per bed, the average length of hospital stay, the number of outpatient visits.

The next step was to use one of the methods to choose the input and output variables (see [76]). The correlation analysis is used to analyze the degree of interdependence of the variables. Based on the achieved values and the authors' recommendations, highly correlated variables are not used in the model (number of doctors, number of nurses, number of physiotherapists, number of treatment days), as they would bring only minimal additional information to the DEA model. Table 2 shows the results of the correlation analysis for the input (3) and output (2) variables, and the variables are described.

**Table 2.** Results of the correlation analysis–inputs and outputs of the DEA model. Source: own processing.

|  | Input_1 | Input_2 | Input_3 | Output_1 | Output_2 |
|---|---|---|---|---|---|
| Input_1 | 1.0000 |  |  |  |  |
| Input_2 | 0.6137 | 1.0000 |  |  |  |
| Input_3 | 0.4412 | 0.5723 | 1.0000 |  |  |
| Output_1 | 0.4740 | 0.5768 | 0.4266 | 1.0000 |  |
| Output_2 | 0.5778 | 0.6265 | 0.4858 | 0.3145 | 1.0000 |

(a) Total number of beds (Input_01)—bed stock of spa facilities, including year-round and seasonal beds, properly equipped with linen and other accessories and complying with medical requirements and regulations.

(b) Total number of employees (Input_02)—total recalculated number of employees working in a spa care facility, regardless their job and classification.

(c) Number of medical staff (Input_03)—the total recalculated number of employees working in a spa care as doctors, nurses, physiotherapists, nurse assistants and nutritionists.

(d) Use of bed capacity (Output_01)—is given by the ratio of the number of treatment days and the actual bed capacity in the number of treatment days, expressed in%.

(e) Number of treated clients (Output_02)—the total number of treated clients (in a calendar year), provided with comprehensive health care in a given spa facility, regardless of the method of payment and the country of origin.

Another important assumption was the fulfillment of the conditions regarding the number of variables to the number of compared DMUs, which state e.g., authors [77]. With regard to the 21 spa companies analyzed, the total number of variables was not allowed to exceed the limit of seven, meaning one-third of the total DMUs analyzed. The condition was met.

## 4. Results

Although the input-oriented DEA model with the assumption of variable returns from scale (BCC-I) is considered to be the starting point in the field of healthcare and hospitality, four variants of input and output-oriented DEA models were applied with the assumption of constant and variable returns from the scale–CCR-I, CCR-O, BCC-I, BCC-O, the most frequently used on the basis of the literature research.

In Table 3, the results are reported also in the case of other basic DEA models most often applied in the area analyzed. The intention is to identify the differences in the results and to monitor the change in the position of the enterprises in the rating.

**Table 3.** Average efficiency scores of the Slovak spa enterprises. Source: own processing.

| Spa Enterprise | Efficiency Scores | | | | Average Ranking | | | |
|---|---|---|---|---|---|---|---|---|
| | CCR-I | CCR-O | BCC-I | BCC-O | CCR-I | CCR-O | BCC-I | BCC-O |
| SE01 | 0.6556 | 0.6556 | 0.7030 | 0.8771 | 12. | 12. | 14. | 13. |
| SE02 | 0.5630 | 0.5630 | 0.5984 | 0.7164 | 13. | 13. | 15. | 18. |
| SE03 | 0.7573 | 0.7573 | 0.8617 | 0.9569 | 6. | 6. | 8. | 8. |
| SE04 | 0.4075 | 0.4075 | 0.5328 | 0.8641 | 19. | 19. | 17. | 14. |
| SE05 | 0.7238 | 0.7238 | 0.7757 | 0.9172 | 9. | 9. | 10. | 9. |
| SE06 | 0.4979 | 0.4979 | 0.8779 | 0.9712 | 15. | 15. | 7. | 7. |
| SE07 | 0.7460 | 0.7460 | 0.7674 | 0.8839 | 7. | 7. | 11. | 12. |
| SE08 | 0.7106 | 0.7106 | 0.7631 | 0.8991 | 10. | 10. | 12. | 10. |
| SE09 | 0.4498 | 0.4498 | 0.4582 | 0.7716 | 17. | 17. | 20. | 17. |
| SE10 | 0.4344 | 0.4344 | 0.5005 | 0.6197 | 18. | 18. | 18. | 21. |
| SE11 | 0.4068 | 0.4068 | 0.4126 | 0.7127 | 20. | 20. | 21. | 19. |
| SE12 | 0.4836 | 0.4836 | 0.4999 | 0.8207 | 16. | 16. | 19. | 15. |
| SE13 | 0.6774 | 0.6774 | 0.7166 | 0.8191 | 11. | 11. | 13. | 16. |
| SE14 | 0.5322 | 0.5322 | 0.5536 | 0.7047 | 14. | 14. | 16. | 20. |
| SE15 | 0.3924 | 0.3924 | 1.0000 | 1.0000 | 21. | 21. | 1. | 1. |
| SE16 | 1,.0000 | 1.0000 | 1.0000 | 1.0000 | 1. | 1. | 1. | 1. |
| SE17 | 0.7704 | 0.7704 | 1.0000 | 1.0000 | 5. | 5. | 1. | 1. |
| SE18 | 1.0000 | 1.0000 | 1.0000 | 1.0000 | 1. | 1. | 1. | 1. |
| SE19 | 1.0000 | 1.0000 | 1.0000 | 1.0000 | 1. | 1. | 1. | 1. |
| SE20 | 0.7418 | 0.7418 | 0.7845 | 0.8927 | 8. | 8. | 9. | 11. |
| SE21 | 1.0000 | 1.0000 | 1.0000 | 1.0000 | 1. | 1. | 1. | 1. |

As reported by the table, through the application of input and output-oriented DEA model with the assumption of constant returns to scale (CCR-I and CCR-O) a total of four enterprises (SE16, SE18, SE19, and SE21) are identified as efficient, based on the achieved average values for 2013–2018. Through the application of input and output-oriented DEA model with the assumption of variable returns from the scale (BCC-I and BCC-O), better results are reported, the model included SE15 and SE17 as efficient as well, increasing the percentage of the efficient enterprises in the Slovak spa industry from 19.05% to 28.57%. Looking at the ranking of the enterprises in the case of application of all the DEA models, the deviations in their locations are minimal, the only significant contrast is identified in the case of SE15. A more-detailed comparison of the results is provided in Figure 1.

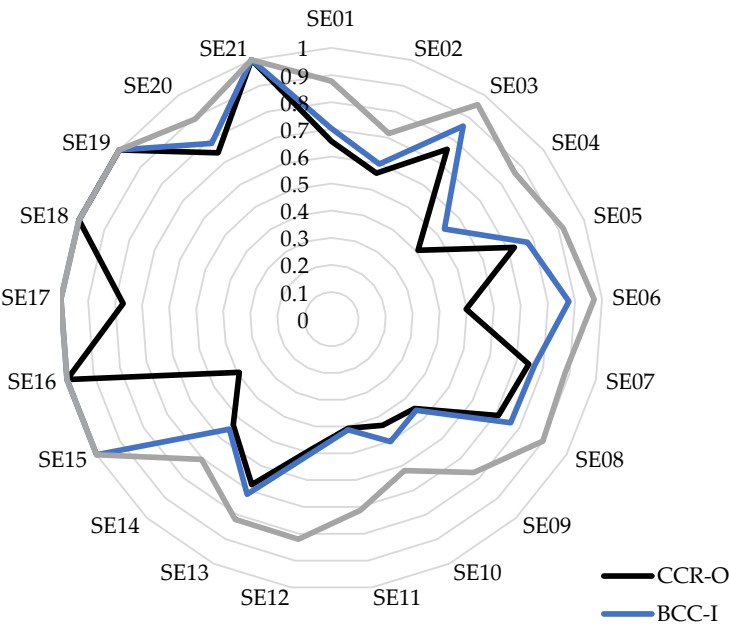

**Figure 1.** Comparison of the efficiency score results–CCR-O, BCC-I, BCC-O DEA models. Source: own processing.

As reported by Figure 1, the average efficiency score is identified identically for the application of all the DEA models only in the case of the efficient enterprises labelled SE16, SE18, SE19, and SE21. In general, the CCR-I and CCR-O models report the same results, so only CCR-O is shown in the graph. Such model evaluated the achieved level of efficiency of the enterprises the most strictly, as evidenced by the lowest average values. The efficiency calculated on the basis of the BCC model is also known as pure technical efficiency, as the BCC model eliminates part of the inefficiency that is caused by the inadequate size of the production unit. It thus divides the efficiency measured by the CCR model into pure technical efficiency and scale efficiency. As further analyses are carried out only on the basis of the results of the input-oriented DEA model with the assumption of variable returns to scale (BCC-I), considered to be the starting point regarding the research sample, Table 4 provides the most important descriptive characteristics.

**Table 4.** Descriptive statistics of efficiency scores (BCC-I DEA model). Source: own processing.

| | Efficiency Scores (BCC-I DEA Model) | | | | | |
|---|---|---|---|---|---|---|
| | **2013** | **2014** | **2015** | **2016** | **2017** | **2018** |
| Min | 0.4483 | 0.4368 | 0.3724 | 0.4120 | 0.4053 | 0.3744 |
| Median | 0.7215 | 0.7293 | 0.7269 | 0.7561 | 0.7195 | 0.6983 |
| Mean | 0.7550 | 0.7573 | 0.7687 | 0.7577 | 0.7503 | 0.7269 |
| Max | 1.0000 | 1.0000 | 1.0000 | 1.0000 | 1.0000 | 1.0000 |
| Standard deviation | 0.2248 | 0.2078 | 0.2132 | 0.2089 | 0.2312 | 0.2189 |

By applying the final BCC-I DEA model, best reflecting the essence of the production process of the spa enterprises, the efficiency scores are quantified, providing a rank rating of the enterprises, identifying their strengths and weaknesses, monitoring of year-to-year efficiency changes, and the identification of the benchmarks. For the inefficient enterprises a projection of reference values of input variables leading to the achievement of the required efficiency limit is provided.

Based on the data, it is clear that the achieved level of efficiency of the spa enterprises did not change significantly during the years. The minimum values ranged from 0.3724 (2015) to 0.4483 (2013). The deviations between the median and the average confirm the absence of extremes in the group, which was one of the prerequisites for the correct application of the DEA method. The development of the average value of technical efficiency showed a positive growing trend until 2015, but in the last three years of the observed period, the efficiency decreased by 1.84% year-on-year. The total average value for all enterprises and years analyzed reached 0.7527, i.e., an average spa enterprise would only need 75.27% of the currently used inputs for a given output production to move to the efficiency limit. The maximum efficiency rate of 1.0000 was achieved in several years by several enterprises, also reflecting the overall development of average efficiency in the sector.

As reported by Figure 2, in each year of the period, the number of inefficient enterprises exceeded the number of the efficient ones; however, their ratio did not change significantly. The highest number of efficient enterprises was achieved in 2013 and 2017, but in the following years there was the most significant decrease. On average, there were seven efficient enterprises (33.33%) and 14 inefficient enterprises (66.67%) in the spa treatment sector. The results of the analyses also point to the fact that during the analyzed years, out of seven efficient spa companies, four operated under conditions of declining scale returns and three enterprises under conditions of constant scale returns.

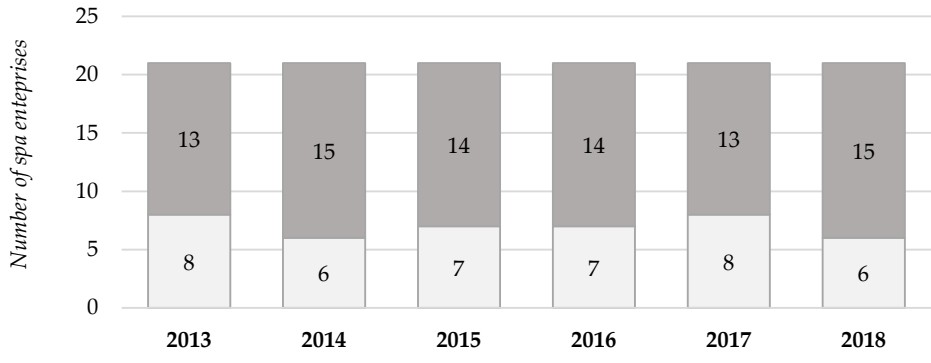

**Figure 2.** Number of efficient and inefficient spa enterprises (2013–2018). Source: own processing.

After the identification of the efficient and inefficient emprises, the compilation of their order on the basis of the achieved degree of efficiency, and the identification of the type of economies of scale, it was possible to proceed to the determination of target, and reference values of the input variables for inefficient units, which is considered to be one of the biggest advantages of the DEA method. Since the BCC-I model is applied, Table 5 presents the real values of the input variables and subsequently their recommended values, which would lead to the achievement of the required efficiency limit. Due to the limited scope of the paper and also the sensitivity of the provided data, the values are not provided separately for each enterprise; it provided the average values for all spa enterprises as a whole.

**Table 5.** Input reference values for reaching the efficiency frontier. Source: own processing.

| Year | Input_01 | | | Input_02 | | | Input_03 | | |
|---|---|---|---|---|---|---|---|---|---|
| | Actual Value | Optimum Value | % Change | Actual Value | Optimum Value | % Change | Actual Value | Optimum Value | % Change |
| 2013 | 541.05 | 417.01 | −24.71 | 188.03 | 143.54 | −27.06 | 55.34 | 40.20 | −32.11 |
| 2014 | 544.14 | 387.62 | −29.58 | 186.97 | 143.82 | −27.32 | 54.03 | 37.69 | −34.32 |
| 2015 | 543.14 | 397.76 | −27.01 | 186.63 | 146.24 | −25.26 | 53.22 | 37.09 | −34.07 |
| 2016 | 539.71 | 356.76 | −32.07 | 187.20 | 145.41 | −26.57 | 52.34 | 35.81 | −34.88 |
| 2017 | 551.48 | 359.18 | −32.33 | 191.48 | 146.90 | −26.95 | 53.25 | 36.15 | −34.42 |
| 2018 | 550.33 | 351.97 | −34.36 | 193.70 | 144.10 | −30.10 | 48.88 | 33.36 | −34.33 |

As the results of previous analyses revealed that there are also inefficient units in the Slovak spa, the projection of reference values was justified in this case. For example, in 2013, the spa enterprises would have to reduce Input_01 (Total number of beds) by 24.71%, Input_02 (Total number of employees) by 27.06%, and Input_03 (Number of medical staff) by 32.11% from the original (actual) values in order to achieve the required efficiency limit (1.0000) subject to the condition of unchanged outputs. In a similar way, the situation in the following years of the analyzed period could be commented, but the situation did not change significantly in the year-to-year comparison. The actual values of the input variables in individual years were provided to us directly by the managers the enterprises, who, however, did not give us consent to their publication. It should be noted that the target values of the inputs were quantified using the vectors of the optimal values of the variables and the input values of the efficient production units.

Finally, based on the achieved average values of efficiency, Table 6 provides a rank rating of the enterprises reflecting the development of their position vis-à-vis other competitors operating in the sector.

**Table 6.** Ranking development of the Slovak spa enterprises (2013–2018). Source: own processing.

| Spa Enterprise | 2013 | 2014 | 2015 | 2016 | 2017 | 2018 | Average Ranking (2013–2018) | Change (2013–2018) |
|---|---|---|---|---|---|---|---|---|
| SE01 | 13. | 12. | 14. | 12. | 11. | 11. | 13. | ↑ 2 |
| SE02 | 14. | 15. | 16. | 17. | 16. | 16. | 15. | ↓ 2 |
| SE03 | 1. | 7 | 8. | 9. | 10. | 13. | 8. | ↓ 12 |
| SE04 | 21. | 19. | 15. | 14. | 19. | 17. | 17. | ↑ 4 |
| SE05 | 9. | 11. | 10. | 10. | 12. | 10. | 11. | ↓ 1 |
| SE06 | 15. | 14. | 1. | 1. | 1. | 7. | 7. | ↑ 8 |
| SE07 | 12. | 8. | 9. | 13. | 13. | 9. | 10. | ↑ 3 |
| SE08 | 10. | 13. | 13. | 8. | 9. | 12. | 12. | ↓ 2 |
| SE09 | 19. | 18. | 20. | 19. | 20. | 21. | 20. | ↓ 2 |
| SE10 | 16. | 17. | 17. | 18. | 17. | 19. | 19. | ↓ 3 |
| SE11 | 20. | 21. | 21. | 21. | 21. | 20. | 21. | ↓↑ 0 |
| SE12 | 17. | 16. | 19. | 20. | 18. | 18. | 18. | ↓ 1 |
| SE13 | 1. | 9. | 11. | 16. | 15. | 15. | 14. | ↓ 14 |
| SE14 | 18. | 20. | 18. | 15. | 14. | 14. | 16. | ↑ 4 |
| SE15 | 1. | 1. | 1. | 1. | 1. | 1. | 1. | ↓↑ 0 |
| SE16 | 1. | 1. | 1. | 1. | 1. | 1. | 1. | ↓↑ 0 |
| SE17 | 1. | 1. | 1. | 1. | 1. | 1. | 1. | ↓↑ 0 |
| SE18 | 1. | 1. | 1. | 1. | 1. | 1. | 1. | ↓↑ 0 |
| SE19 | 1. | 1. | 1. | 1. | 1. | 1. | 1. | ↓↑ 0 |
| SE20 | 11. | 10. | 12. | 11. | 1. | 8. | 9. | ↑ 3 |
| SE21 | 1. | 1. | 1. | 1. | 1. | 1. | 1. | ↓↑ 0 |

In 2013, there were eight spa enterprises ranked the first (SE03, SE13, SE15, SE16, SE17, SE18, SE19, SE21), as all of them report PPF and achieved an efficiency rate of 1.0000. In the following year, their number decreased to six effective units (SE15, SE16, SE17, SE18, SE19, SE21), in the case of SE03 the efficiency score decreased to 0.9863, mainly due to a disproportionate increase in Input_01. The identical development caused a decrease in the efficiency of the SE06 (0.6922), which could thus use approximately 30.78% less inputs in the year and thus reach the required efficiency limit. However, the situation improved again in the following years 2015–2017, SE06 and in 2017 also SE20 joined the stable leaders of the given area. In 2017, the first place was occupied again by eight spa enterprises; however these enterprises did not manage to stabilize their position in the following year.

Regarding the change in the position of the enterprises over time, no shift was recorded in the case of SE11, SE15, SE16, SE17, SE18, SE19, and also SE21. With the exception of SE11, the remaining group of the efficient enterprises is identified as a model reference group for other inefficient enterprises in the sector and thus be able to help them manage their resources (inputs) more rationally. As a result of the positive development in the years 2015 to 2017, the already mentioned company KP06 significantly improved its overall position, and on the contrary, the most significant drop was recorded in the case of SE13 and SE03.

## 5. Discussion

As mentioned in the previous text, the spa treatment facilities are rather specific business entities and so far they have not been the subject of any research study evaluating their effectiveness by applying any DEA model, not yet in Slovakia. For this reason, it was not possible to make any comparison of the results achieved on the development and current state of efficiency of Slovak spa enterprises with other research. However, the issue of the Slovak spa industry is addressed, for example by [78–81], their research is focused more on the evaluation of the financial situation and performance of the spa enterprises and the economic impact of the health insurance system on Slovak spas, rather than on the efficiency analyses. Therefore this research study is pioneering and beneficial for practice, not only in the Slovak Republic.

Due to the facts as mentioned above, the discussion focuses on the comparison of the overall construction and selection of the variables in the DEA model as presented in the paper and world and Slovak empirical studies carried out previously. The activities of the spa enterprises intersect in the field of hospitality and healthcare, although the priority attention should, in our opinion, be focused on the provision of spa healthcare. Due to the limited scope of the paper, Table 7 reports the most important world empirical studies carried out in medical facilities. The comparison focuses on comparing the selection of the input and output variables that significantly affect the relevance of the results achieved.

**Table 7.** Overview of inputs and outputs in DEA models (healthcare facilities). Source: own processing.

| Authors | Country | Inputs | Outputs |
|---|---|---|---|
| [81] | Greece | Number of doctors and nurses in the hospital | Number of medical examinations, laboratory tests and transfers from medical centers to hospitals |
| [70] | Czech Republic | Amount of operating costs | Number of beds, number of hospitalized patients, use of bed (in days) |
| [49] | Hungary | Number of beds, doctors, nurses and other professional staff | Number of days spent by patients on the ward, number of discharged patients |

| | | | |
|---|---|---|---|
| [82] | Poland | Average length of hospital stay (in days), average cost of daily hospital care | Average number of patients per bed, share of accredited hospitals in total, net annual income of the doctor |
| [83] | Iran | Number of doctors, number of nurses, number of active beds and facilities | Bed occupancy rate, number of patients discharged, price per bed, doctors' fees |
| [56] | Italy | Number of beds, number of employees, number of doctors, nurses and other medical staff, operating costs | Number of hospital days, number of outpatient visits |
| [84] | China | Number of beds, number of medical technicians | Sales, number of discharged patients, number of outpatient visits |

Analysis of the effectiveness of medical facilities (mostly hospitals) has been the subject of research studies by several Slovak authors. As part of the analysis of inputs and outputs of different DEA models, very similar (in some cases even identical) indicators were used as in the case of the above-mentioned world studies. For example authors [69] include the number of beds, doctors, nurses, and other health care workers in the input variables. The number of patients and the number of days spent in the hospital are used as the output variables. The inputs within the DEA model by Sendek et al. [68] consist of the number of beds, bed costs, and the cost of drugs and medical products. On the other hand, the number of hospitalizations and outpatient visits to hospitals represent the output variables of the model. Szabo et al. [85] analyze the effectiveness of the Slovak medical facilities using inputs such as the number of employees, the number of beds, the amount of material, and labor costs. The outputs of the model are represented by the number of hospitalizations, operations, and also the number of outpatient visits. For a more comprehensive comparison of the relevant design of our DEA model, Table 8 offers an overview of the input and output variables within the DEA models applied in hotel facilities.

**Table 8.** Overview of inputs and outputs in DEA models (hotel facilities). Source: own processing.

| Authors | Country | Inputs | Outputs |
|---|---|---|---|
| [86] | Spain | Assets, material costs, labor costs | EBIT |
| [57] | Croatia | Energy costs, costs per room, food and beverage costs, costs associated with other services, labor costs | Sales, occupancy rate |
| [65] | Spain | Number of employees, labor costs, number of rooms | Sales, amount of sales per room, market share |
| [50] | Greece | Number of local units, number of employees, investments | Sales |
| [87] | Italy | Tangible assets, intangible assets, labor costs | Sales |
| [88] | Spain | labor costs, depreciation, operating costs | Sales |
| [63] | Ecuador | Number of employees, fixed assets, consumption costs | Sales |

Many authors in the Slovak Republic have not discussed the efficiency of hotels using the DEA method. Gúčik and Uličná [89] analyze the dynamics of efficiency development using the Malmquist index on a sample of 50 hotels in the years of 2010–2013, including the number of rooms and average labor costs in the input variables. They use exclusively the total annual revenues of hotels as the output variable. Horváthová and Mokrišová [51] use the financial indicators in the inputs and outputs when evaluating the sample of 25

hotels. However, their research is not focused primarily on evaluating the technical efficiency of Slovak hotels, but rather on examining the possibility of using the DEA method as an alternative to Altman's model of predicting the risk of corporate bankruptcy.

## 6. Conclusions

Measuring efficiency and identifying the sources of potential inefficiency in particular are very important steps in improving the competitive position of the enterprises in their continuous development, sustainability, overall behavior in the current corporate environment and security aspects [36]. An exception is not made by the emprises operating in the progressively developing field of tourism–spa treatment. At present, the importance of spa tourism is constantly growing, due to the growing awareness of people in the care of their own health and healthy lifestyle. It is of great importance for both domestic and inbound tourism and, like other forms of tourism, it is an important source of income for the private, municipal, and public sectors. Nevertheless, in our opinion, not enough attention is paid to it, which is directly linked to their support and sustainable development.

The paper focuses on evaluating the overall development and current level of efficiency of the Slovak spa sector in the years of 2013–2018 through the application of the DEA models. Despite the fact that in the case of all DEA models as applied in the paper, the deviations of the achieved efficiency score are minimal, the most relevant and practically applicable results were reported for the spa enterprises analyzed using the input-oriented DEA model with variable range of returns (BCC-I). The argument can be justified in particular by the fact that the spa facilities in question have a social responsibility for the provision of medical treatment and public care, while the influence of inputs in the area is more permissible and realistic compared to the outputs. At the same time, it is not possible to assume constant returns from scale in the sense of a linear increase in the outputs with the increasing inputs and vice versa. The assumption of constant returns to scale is accepted if all DMUs operate at the optimum size. Imperfect competition, regulatory, financial constraints, and other factors make this assumption much impossible.

Based on the results of the BCC-I DEA model, the achieved efficiency of the spa enterprises did not change significantly during the analyzed years. The development of the average value of technical efficiency showed a positive growing trend until 2015, but in the last three years it decreased by 1.84% year-to-years. The total average value for all the enterprises and analyzed years reached the amount of 0.7527, i.e., the average spa enterprise would need only 75.27% of currently used inputs for a given output production to move to the efficiency limit. In each year of the analyzed period, the number of inefficient companies exceeded the efficient ones, but their ratio did not change significantly—on average, seven efficient enterprises (33.33%) and 14 inefficient enterprises (66.67%) operated in the spa treatment area. This research study is beneficial both for the managers of the Slovak spa enterprises, and also for the professional academic public.

In terms of managerial implications, the research results are applicable in the Slovak spa sector. The practical aspect monitors factors such as the size of the business and the current situation of the spa. The definition of the market position of the spa company and the comparison with other competitors on the market is significant for managerial benefit. The results of the research are applicable not only to Slovak tourism SMEs, but also in the surrounding countries in the region. These countries have undergone similar economic developments and the model is able to analyze these factors.

Due to the specificity and attractiveness of the sample of the spa enterprises, this research study is pioneering and filling the identified gap in the previous empirical studies. Of course, it is important to point out some of the limits and limitations of the paper. One of the most significant limits is the size of the research sample. As there are only a total of 30 spa enterprises operating in Slovakia with the official permit of the Ministry of Health of the Slovak Republic, it was not possible to influence the given fact in any way. Another

limitation of the study is the limitation of the provided data in the case of variables entering the DEA model. Not all analyzed spa enterprises keep the records of the data required by us, and it was necessary to exclude them from further analyses due to incomplete data. The achieved efficiency score depended exclusively on predefined inputs and outputs. In the case of selecting the other variables (e.g., financial) in combination with another type of selected DEA model, other results would probably be achieved, which, however, goes beyond the scope of this paper, but also creates space for future research perspectives. We plan to expand the research sample to include the enterprises from other European countries and to monitor how the position of the Slovak spa industry will change, as well as the ranking of different spa enterprises in the context of a global comparison.

**Author Contributions:** Conceptualization, V.Č., J.D., and P.G.; methodology, V.Č. and P.G.; software, V.Č., P.G., and B.B.; validation, J.D., P.G., and J.V.; formal analysis, P.P., J.V., and B.B.; investigation, P.G., J.O., and V.Č.; resources, V.Č., P.P., and B.B.; data curation, J.D., P.G., and J.V.; writing—original draft preparation, V.Č., B.B., J.O., and P.P.; writing—review and editing, P.G., J.D., and P.P.; visualization, V.Č. and B.B.; supervision, V.Č., P.G., and J.D.; project administration, J.D.; funding acquisition, P.P. and J.V. All authors have read and agreed to the published version of the manuscript.

**Funding:** This research received no external funding.

**Institutional Review Board Statement:** Not applicable.

**Informed Consent Statement:** Not applicable.

**Data Availability Statement:** Not applicable.

**Conflicts of Interest:** The authors declare no conflict of interest. The funders had no role in the design of the study; in the collection, analyses, or interpretation of data; in the writing of the manuscript, or in the decision to publish the results.

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
