# Peer review of "Application of the DEA Model in Tourism SMEs: An Empirical Study from Slovakia in the Context of Business Sustainability"

_sustainability, doi:10.3390/su13137422_

Round 1
Reviewer 1 Report
Dear Authors,
I find your paper very interesting, however, there are some issues:
1) there are some problems with references (sometimes bold, sometimes page numbers); check line 93 - please try to use Mendeley and there is a Sustainability journal style and this can help you;
2) names of the companies could be listed in the appendix section;
3) The title of your paper is too long, can you please rethink it?
4) line 38 - please write Data form big letter in your keyword;
5) Why do you use color in line 570 ?
6) can you please add some managerial implications to your work? Maybe spas in Slovakia gain Sustainable competitive advantage as Soloducho-Pec (2020) suggests?
In your reference section not always all letters have to be written in capital letters, check your references and try to avoid writing whole sentences or country names in big letters (line 657)
I find this paper well written with a little number of mistakes, which are generally more technical than scientific, and my general opinion is positive
Author Response
Dear reviewer,
we really appreciate your inspiring comments and recommendations as they have contributed to the improvement of the quality of our research paper.
- Errors in citing the bibliographic sources were adjusted according to the official template. Some references were used several times in the text (Farell 1957, Dlouhý et al. 2018, Charnes et al. 1978, Dénes et al. 2017, Karakitsiou et al. 2018, Horváthová and Mokrišová 2018, Hajduová et al. 2015, Samut and Cafri 2016, Poldrugovac et al. 2016, Mendieta-Peñalver et al. 2016, Higuerey et al. 2020, Lacko et al. 2015, Papadaki and Staňková 2016, Szabo et al. 2018, Sendek et al. 2015, Ivančík and Nečas 2017) and thus their order in References was not maintained.
- We originally planned to include the names of the spas in the appendices, but in the end we decided to keep them in the main text of the article.
- The title of the paper was slightly shortened and modified.
- Keyword "data Envelopment Analysis" was corrected (first capital letter).
- Formal/typing error in line 570 was corrected.
- A short paragraph about managerial implications of our research was added.
Reviewer 2 Report
- The abstract must be restructured in accordance with the technical editing requirements of the journal (it is large).
- At the end of the Introduction section, I think you need to add a paragraph in which you emphasize the novelty of the research, the usefulness of the research.
- Page 2 lines 90 and 92: there are errors in citing the bibliographic sources.
- Avoid using the personal pronoun (we, our)
- The paper states that the strengths and weaknesses of spa businesses are identified. What is the action plan to minimize weaknesses?
- What are the managerial implications generated by the application of DEA methods?
- I think the conclusions section should present the results of the research much better. Also add research limits, future research perspectives.
- Page 16 line 632: typing error.
Author Response
Dear reviewer,
we really appreciate your inspiring comments and recommendations as they have contributed to the improvement of the quality of our research paper.
- The abstract of the paper was slightly modified and shortened according to the recommendations; its structure was verified.
- A short paragraph emphasizing the novelty and usefulness of the research was added.
- Errors in citing the bibliographic sources were corrected.
- The use of personal pronouns in the text was checked and corrected (removed).
- A short paragraph on how to minimize the identified weaknesses of the Slovak spa sector was added.
- A short paragraph about managerial implications of our research was added.
- Our intention was to include as many results as possible in the paper, but probably at the expense of easier understanding of the text. According to your recommendation, the presentation of the results was adjusted and text was divided into more paragraphs. Also, a short paragraph on research limits as well as future research direction was added (removed from the previous section).
- Typing error in line 632 was corrected.
Reviewer 3 Report
This study considered total number of beds total number of employee and number of medical staff as key factors to decide to the efficiency of spa section.
To investigate the efficiency of spa in Slovakia, I recommend the profit or revenue is included in main input rather than number of medical staff.
Moreover, service score of customer should be considered in output variable.
Author Response
Dear reviewer,
we really appreciate your recommendations. Application of the DEA model using the financial variables was the subject of our previous research studies (e. g. Čabinová, V.; Onuferová, E.; Fedorčíková, R.; Sikorová, N. Efficiency Evaluation of the Medical Spa Sector in Slovakia: An Application of DEA Method). This time, we have focused only on the non-financial variables. Journal of Management Systems 2020, 21, 7–14). However, the combination of the financial and non-financial DEA inputs/outputs within the efficiency evaluation of Slovak spa enterprises is one of our further research intentions.
Round 2
Reviewer 2 Report
The article has been significantly improved.
Author Response
We added (red):
- Introduction
Due to changing global business environment, the issue of evaluating business efficiency is currently discussed more, not to mention the effort to develop new, modern approaches to its solution. Due to the development of statistical methods and the growing interest of financial institutions and the enterprises, there is a significant increase in the use of innovative methods and the models for evaluating business efficiency in various economic sectors, particularly in the last decade. It also applies to the progressively evolving field of tourism, as its support and balanced sustainable development began to play a crucial role in the national economies [1]. The research sample is based on current demographic trends, using a specific and attractive research sample – the spa enterprises. Spa tourism is one of the most popular forms of recreation in the modern world. In the last three decades, there has been a dynamic development of spa tourism and other forms of health tourism, such as wellness and medical tourism [2,3]. By the medium variant of the United Nations forecast calculated for the countries of the European Union, it is expected that almost 29% of the population will be over 65 years of age by 2050, providing space for promising clients. Compared to other European Union member states, there is a growing interest in business activity in Slovakia [4]. The business sector is undoubtedly one of the most important parts of today's market economy. SMEs produce and sell the major part of goods and services, thus contributing to the sales tax revenues, GDP and job creation. In this regard, their role in the economy is irreplaceable and justified [5]. However, during pandemic, enterprises face pressure and are forced to deal with many unexpected problems and challenges in order to ensure sustainable entrepreneurship [6]. Undoubtedly, tourism is the engine of economic growth, as reflected in foreign exchange earnings; contributions to private and public revenues; in job creation; incentives for technology development; in the formation of human capital, business opportunities, etc. [7,8,9]. To the above, together with other reasons, it is therefore important to pay attention to such form of tourism, assess its current state and focus on its support and sustainable development.
................................................................................................................................................................................................................................................................................................................................................................................
References
[4]. Mura, L.; Hajduová, Z. Measuring Efficiency by Using Selected Determinants in Regional SMEs, Entrepreneurship and Sustainability Issues 2021, 8, 487–503.
[5]. Fiľa, M.; Levicky, M.; Mura, L.; Maros, M.; Korenkova, M. Innovations for Business Management: Motivation and Barriers. Marketing and Management of Innovations 2020, 4, 266–278.
[6]. Mura, L. Marketing Management of Family Businesses: Results of Empirical Study. International Journal of Entrepreneurial Knowledge 2020, 8, 56–66.
Reviewer 3 Report
Most comments were considered in revised version.
Author Response

(The authors gave the same response as above.)
